# Observation of non-Hermitian topological Anderson insulator in quantum dynamics

Quan Lin[1,4], Tianyu Li[2,3,4], Lei Xiao[1], Kunkun Wang[1], Wei Yi [2,3✉] & Peng Xue [1✉]

Disorder and non-Hermiticity dramatically impact the topological and localization properties of a quantum system, giving rise to intriguing quantum states of matter. The rich interplay of disorder, non-Hermiticity, and topology is epitomized by the recently proposed non-Hermitian topological Anderson insulator that hosts a plethora of exotic phenomena. Here we experimentally simulate the non-Hermitian topological Anderson insulator using disordered photonic quantum walks, and characterize its localization and topological properties. In particular, we focus on the competition between Anderson localization induced by random disorder, and the non-Hermitian skin effect under which all eigenstates are squeezed toward the boundary. The two distinct localization mechanisms prompt a non-monotonous change in profile of the Lyapunov exponent, which we experimentally reveal through dynamic observables. We then probe the disorder-induced topological phase transitions, and demonstrate their biorthogonal criticality. Our experiment further advances the frontier of synthetic topology in open systems.

[1] Beijing Computational Science Research Center, Beijing 100084, China. [2] CAS Key Laboratory of Quantum Information, University of Science and Technology of China, Hefei 230026, China. [3] CAS Center For Excellence in Quantum Information and Quantum Physics, Hefei 230026, China. [4] These authors contributed equally: Quan Lin, Tianyu Li. ✉email: wyiz@ustc.edu.cn; gnep.eux@gmail.com

Topological edge states in topological materials are robust against weak perturbations, an ability originating from the global geometry of eigen wave functions in the Hilbert space[1,2]. Such an intrinsic geometric feature is captured by global topological invariants that are related to edge states through the bulk-boundary correspondence. However, this conventional paradigm is challenged by localization under disorder[3–6] or non-Hermiticity[7–24], which have become the focus of study of late, particularly in light of recent experimental progress in synthetic topological systems[25–35]. On one hand, despite its gap-closing tendency, the disorder can induce topology from a trivial insulator. In the resulting topological Anderson insulator, the global topology emerges in a bulk with localized states, in the absence of translational symmetry[3–6]. On the other hand, in a broad class of non-Hermitian topological systems, the nominal bulk states are exponentially localized toward boundaries under the non-Hermitian skin effect[8–22]. The deviation of the bulk-state wave functions from the extended Bloch waves invalidates the conventional bulk-boundary correspondence, necessitating the introduction of non-Bloch topological invariants[8–11]. While the two localization mechanisms differ in origin and manifestation, the topology of the underlying system gets fundamentally modified in either case. Remarkably, in the recently proposed non-Hermitian topological Anderson insulator[36–39], the two distinct localization mechanisms are pitted against each other, wherein the interplay of disorder, non-Hermiticity, and topology leads to exotic phenomena such as the non-monotonous localization, disorder-induced non-Bloch topological phase transitions, and biorthogonal critical behaviors.

In this work, we report the experimental observation of non-Hermitian topological Anderson insulators in single-photon quantum-walk dynamics. Driven by a non-unitary topological Floquet operator, the quantum walk undergoes polarization-dependent photon loss and acquires the non-Hermitian skin effect. In contrast to previously implemented quantum walks with the non-Hermitian skin effect[30,35], our current experiment resorts to the time-multiplexed configuration, with the spatial degrees of freedom encoded in the discrete arrival time of photons at the detector[40]. This enables us to implement quantum walks with a larger number of time steps, which is pivotal for the current experiment. We introduce static random disorder through parameters of the optical elements[41], which would result in a complete localization of bulk states in the large-disorder limit. In the intermediate regime with moderate loss and disorder, the competition between the non-Hermitian skin effect and Anderson localization yields non-monotonic localization features which we characterize by measuring the Lyapunov exponent[20]. Using the biorthogonal chiral displacement, we then probe the topological phase transition, which is in qualitatively agreement with theoretical predictions. At the measured topological phase boundary, the biorthogonal localization length diverges, consistent with the biorthogonal critical nature of the phase transition[36–38]. We further measure topological edge states from dynamics close to the boundary of the non-Hermitian topological Anderson insulator.

## Results

**A time-multiplexed non-unitary quantum walk**. We implement a one-dimensional photonic quantum walk governed by the Floquet operator

$$U = R(\theta_2)MSR(\theta_1)MSR(\theta_2). \tag{1}$$

Here the shift operator is given by $S = \sum_x |x-1\rangle\langle x| \otimes |H\rangle\langle H| + |x+1\rangle\langle x| \otimes |V\rangle\langle V|$, with $|H\rangle$ ($|V\rangle$) the horizontally (vertically)

polarized state. The non-unitary operator $M = \sum_x |x\rangle\langle x| \otimes \begin{pmatrix} e^\gamma & 0 \\ 0 & e^{-\gamma} \end{pmatrix}$ with $\gamma$ the gain-loss parameter. The coin operator $R(\theta) = \sum_x |x\rangle\langle x| \otimes \begin{pmatrix} \cos\theta & -\sin\theta \\ \sin\theta & \cos\theta \end{pmatrix}$, where the matrix is in the basis $\{|H\rangle, |V\rangle\}$. For the quantum-walk dynamics, $U$ is repeatedly acted upon the walker state, giving rise to discrete-time Floquet dynamics. The quantum walk governed by $U$ features the non-Hermitian skin effect (see Supplemental Material), which originates from a non-vanishing bulk probability flow that we confirm later with dynamic measurements.

For the experimental implementation, we adopt a time-multiplexed scheme, as illustrated in Fig. 1. Photons are sent through an interferometric network consisting of optical elements for a half step of the discrete-time quantum walk in Eq. (1). The shift operator is implemented by separating the two polarization components and routing them through fibers of different lengths to introduce a polarization-dependent time delay, such that the walker position is mapped to the time domain. For instance, a superposition of multiple spatial positions at a given time step is translated into the superposition of multiple well-resolved pulses within the same discrete-time step. A pair of wave plates are introduced into each of the paths, to realize a polarization-dependent loss operation $M_E = \sum_x |x\rangle\langle x| \otimes (|H\rangle\langle H| + e^{-2\gamma}|V\rangle\langle V|)$, which is related to $M$ through $M = e^\gamma M_E$. We, therefore, read out the time-evolved state driven by $U$ by adding a time-dependent factor $e^{\gamma t}$ to our experimental measurement. To implement the coin operator, an electro-optical modulator (EOM) is inserted into the main interferometric cycle, in combination with wave plates, to provide a carefully time-sequenced control over $\theta$. Importantly, the EOM enables an individual-pulse-resolved coin operation, providing the basis for the implementation of a walker-position-dependent disorder. The disorder is introduced to the operator $R(\theta_1)$ in Eq. (1), where the actual rotation angle is modulated by a small position-dependent $\delta\theta(x)$, with $\delta\theta(x)$ randomly taking values within the range of $[-W, W]$. Here $W$ indicates the disorder strength. We implement only static disorder for our experiments, such that $\delta\theta(x)$ does not change with time steps.

For the input and out-coupling of the interferometric network, a beam splitter (BS) with a reflectivity of 5% is introduced, corresponding to a low coupling rate of photons into the network, but also enabling the out-coupling of photons for measurement. For that purpose, two avalanche photodiodes (APDs) are employed to record the out-coupled photons' temporal and polarization properties, yielding information regarding the number of time steps, as well as the spatial and coin states of the walker.

**Non-Hermitian skin effect**. Whereas the non-Hermitian skin effect is typically associated with non-reciprocity[8], it can also occur in systems with on-site loss[7,30]. Here the non-Hermitian skin effect is a result of the interplay of an on-site, polarization-dependent loss ($M_E$ operator) and an effective coupling between the polarization and spatial modes ($S$ operator). While a defining signal of the non-Hermitian skin effect is the accumulation of Eigen wave functions at the boundary, it also impacts dynamics in the bulk, leaving unique signatures in the Lyapunov exponent. Here the Lyapunov exponent is defined as $\lambda(v) = \lim_{t\to\infty} \frac{1}{t}\log|\psi(x = vt, t)|[20]$, where $v$ is the shift velocity, and $\psi(x, t)$ is the wave-function component at position $x$ and time step $t$. Remarkably, for a system with the non-Hermitian skin effect, $\lambda(v)$ takes a maximum value at $v \neq 0$ for bulk dynamics far away from any boundary[20]. By contrast, in the absence of the non-Hermitian skin effect, $\lambda(v)$ acquires a symmetric profile with respect to its peak at $v = 0$. Intuitively, from the

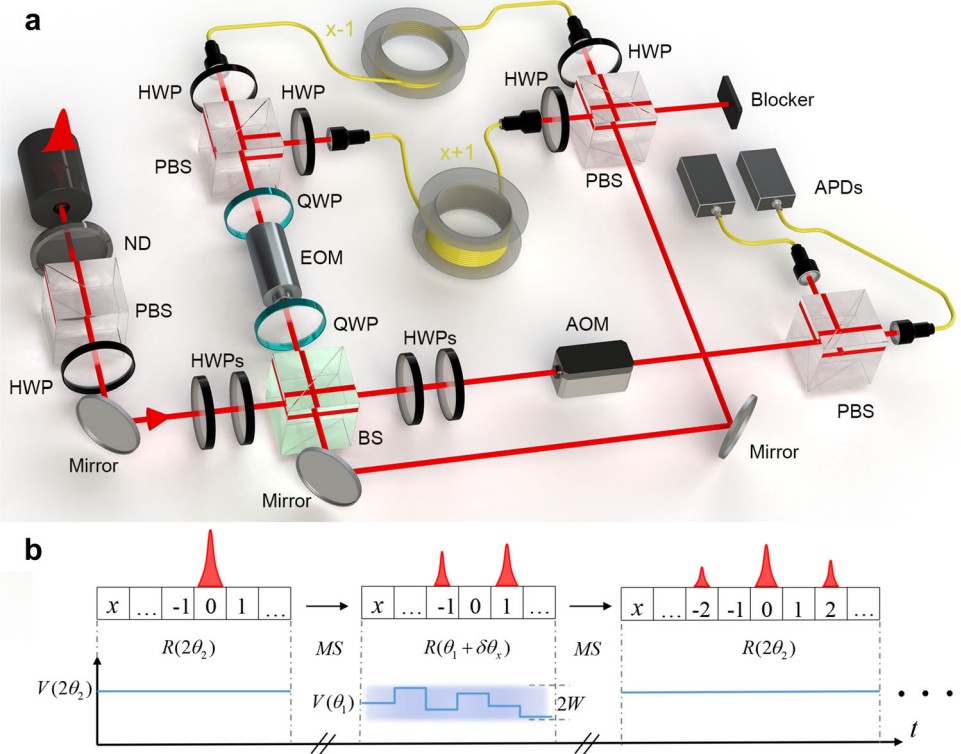

**Fig. 1 Experimental setup for observing non-Hermitian topological Anderson insulator. a** Photons are coupled in and out of an interferometric network through a low-reflectivity beam splitter (BS, reflectivity 5%). The coin operation is carried out with wave plates and a dynamic electro-optic modulator (EOM). The shift operator is realized by splitting the light, using a polarizing beam splitter (PBS), into two single-mode fibers of length of 160,000 and 167,034 m, respectively. As such, the spatial modes are encoded into the polarization-dependent temporal shift within a time step. The out-coupled photons are detected using the avalanche photodiodes (APDs), in a time- and polarization-resolved fashion. An optical switch acousto-optic modulator (AOM) is used to protect the APDs such that photons are only allowed to reach the APDs at the time of measurement. **b** Illustration of the operation sequence of the time-multiplexed quantum walk. Here $V(\theta_i)$ is the control voltage applied to the EOM for generating rotations with the coin parameter $\theta_i$.

definition of the Lyapunov exponent, it is understood that, if $\lambda(v)$ peaks at a shift velocity $v_m$ at time $t$, the time-evolved wave function must peak at $x = v_m t$. A finite peak shift velocity thus reflects a directional wave function propagation in the bulk (or equivalently, a persistent bulk current), which lies at the origin of the non-Hermitian skin effect. Alternatively, the non-Hermitian skin effect can also be confirmed by dynamics close to a boundary (see Supplemental Material).

For our experiment, we implement ten-step quantum walks without imposing any boundary or domain and measure the polarization-averaged growth rate

$$\bar{\lambda}(v) = \frac{\lambda_H(v) + \lambda_V(v)}{2}. \quad (2)$$

Here the additional average over polarization enables us to qualitatively capture the distinctive features of the Lyapunov exponent using a relatively small number of time steps ($t = 10$). In Eq. (2), the polarization-resolved growth rates are defined as $\lambda_i(v) = \frac{1}{t} \log |\psi^{(i)}_{x=vt}|$. To construct $\psi^{(i)}_{x=vt} = \langle i| \otimes \langle x| U^t |0\rangle \otimes |i\rangle$ ($i = H, V$), we initialize the walker in the state $|0\rangle \otimes |i\rangle$, and projectively measure the probability distribution of photons in the polarization state $|i\rangle$ of the spatial mode $|x\rangle$, following the last time step ($t = 10$). Note that the average over polarization in Eq. (2) is taken for faster convergence of the growth rate at a finite evolution time to the Lyapunov exponent.

In Fig. 2, we show the measured polarization-averaged growth rates as functions of the shift velocity, for a, c the unitary, and b, d the non-unitary cases, both without the disorder. Apparently, under the non-Hermitian skin effect ($\gamma \neq 0$), the peak of the

growth rate lies with a finite $v$ (Fig. 2b), in contrast to the more symmetric profile without skin effect (Fig. 2a). Such a growth-rate profile directly originates from the directional propagation of probability in the bulk, as clearly indicated in the measured polarization-resolved probability distributions after the final time step (Fig. 2c, d). In the presence of open boundaries, the directional probability propagation naturally leads to the accumulation of population at the boundaries. Note that the ability to infer the existence of the non-Hermitian skin effect from bulk dynamics confirms that the non-Hermitian skin effect is not merely a finite-size effect, but has a profound impact even within the thermodynamic limit.

**Competition with Anderson localization.** We now switch on disorder and investigate the interplay between the non-Hermitian skin effect and disorder[36,37]. In Fig. 3, we show the measured $\bar{\lambda}(v)$ for increasing disorder strength $W$, under a fixed non-Hermitian parameter $\gamma$. When $W$ is small, the asymmetric profile persists (see Fig. 3a, d), indicating the dominance of the non-Hermitian skin effect. A careful comparison between Fig. 2b and Fig. 3a suggests the emergence of another peak at $v = 0$, though only just visible in Fig. 3a. The peak at $v = 0$ rapidly rises with increasing $W$. This leads to the twin-peak structure under an intermediate $W$, as shown in Fig. 3b and **e**. This is a direct evidence for the competition between the disorder-induced Anderson localization and the non-Hermitian skin effect. Finally, for sufficiently large $W$, $\bar{\lambda}(v)$ again peaks at $v = 0$, as Anderson localization completely suppresses probability flow in the bulk that leads to the non-Hermitian skin effect. Such a competition as revealed by our

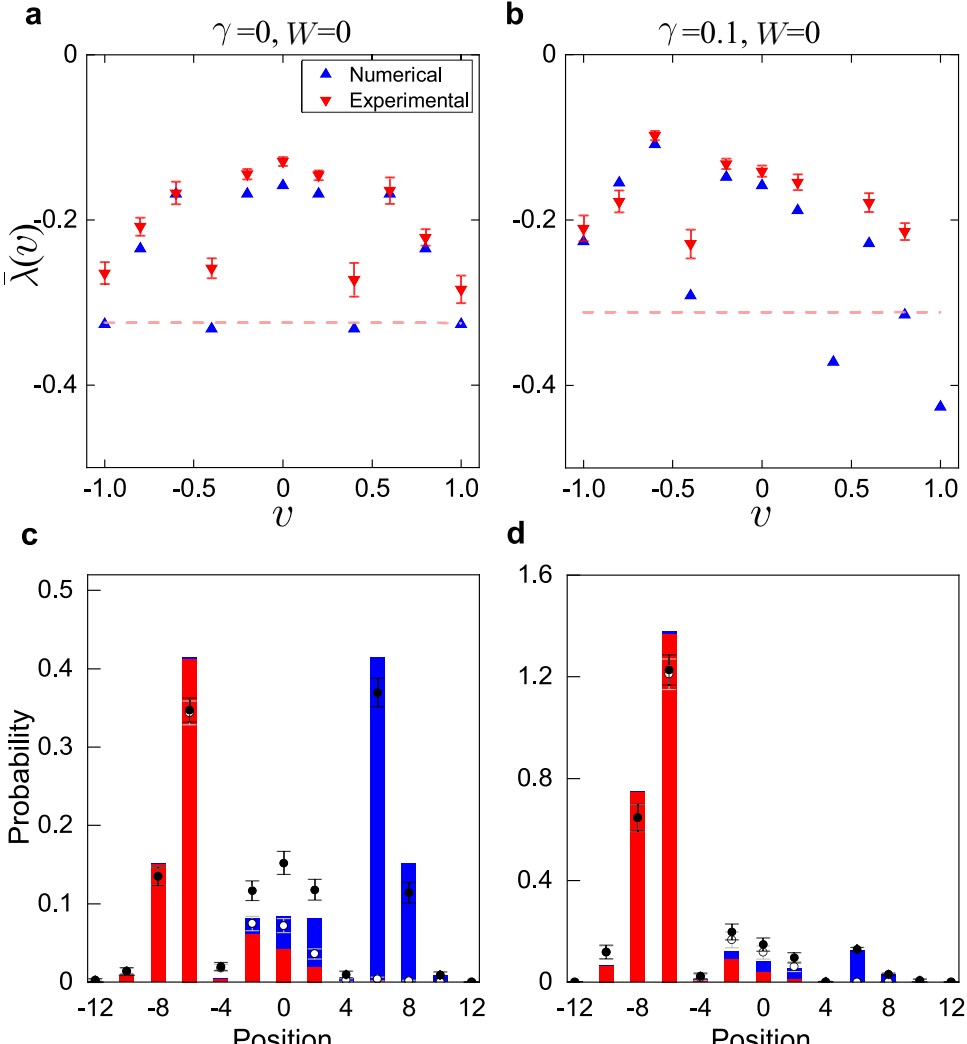

**Fig. 2 Lyapunov exponent from bulk dynamics.** Measured polarization-averaged growth rates $\bar{\lambda}(v)$ for a unitary quantum walk with $\gamma = 0$ in (**a**) and a non-unitary quantum walk with $\gamma = 0.1$ in (**b**). Red triangles with error bars are the experimental data and blue triangles are from numerical simulations. The horizontal dashed line indicates the threshold values below which experimental data were no longer reliable due to photon loss. To construct $\bar{\lambda}$, we initialize the walker in the state $|0\rangle \otimes |H\rangle$ ($|0\rangle \otimes |V\rangle$), evolve it up to ten steps under the parameters ($\theta_1 = 4.3, \theta_2 = 2.175, W = 0$), and projectively measure the horizontally (vertically) polarized photon distribution following the last time step. Note that the system is in a topologically non-trivial phase under the chosen parameters. We construct $\lambda_H$ and $\lambda_V$ from these polarization-resolved distributions, from which $\bar{\lambda}$ is calculated. **c, d** The polarization-resolved photon distribution after the last time step $t = 10$, for the dynamics in **a** and **b**, respectively. For each bar, the blue (top) and red (bottom) portions are respectively the numerical results for the horizontal-polarization-resolved and vertical-polarization-resolved photon distributions, initialized in $|0\rangle \otimes |H\rangle$ and $|0\rangle \otimes |V\rangle$, respectively. The white dots are the experimental measurements for the vertical-polarization-resolved photon distribution, and black dots are the experimental results for the sum of the polarization-resolved distributions. Error bars are due to the statistical uncertainty in photon-number-counting.

experiment is consistent with the non-monotonous localization predicted in ref. [36], where the inverse participation ratio is used to characterize the competition (see Supplemental Material).

**Disorder-induced topology.** The Floquet operator $U$ is topological, protected by the chiral symmetry with $\Gamma U T = U^{-1}$, where $\Gamma = \sum_x |x\rangle\langle x| \otimes \sigma_x$. While the topology of $U$ generally persists under small random disorder, the disorder can also induce non-trivial topology from a topologically trivial state, similar to the case with the topological Anderson insulator in Hermitian systems[3–6]. We emphasize that the topology discussed here is to be differentiated from the spectral topology of the non-Hermitian skin effect, with the latter indicating closed-loop structures of the eigenenergy spectra on the complex plane[17,18].

In Fig. 4a, we plot the theoretical phase diagram, characterized through the disorder-averaged local marker under the non-Bloch band theory (see Supplemental Material). The yellow (blue) region corresponds to the topologically non-trivial (trivial) phase, thus the non-Hermitian topological Anderson insulator state corresponds to the yellow region with the finite disorder ($W > 0$). Here the biorthogonal local marker, calculated over a unit cell deep in the bulk, plays the role of a topological invariant in the presence of disorder and converges to the non-Hermitian winding number for $W = 0$ (see Methods). Incidentally, for our choice of $U$, the topological phase boundary is insensitive to $\gamma$, despite the presence of the non-Hermitian skin effect and the application of the non-Bloch band theory. Nevertheless, the biorthogonal localization length, rather than the conventional localization length, diverges at the topological phase boundary

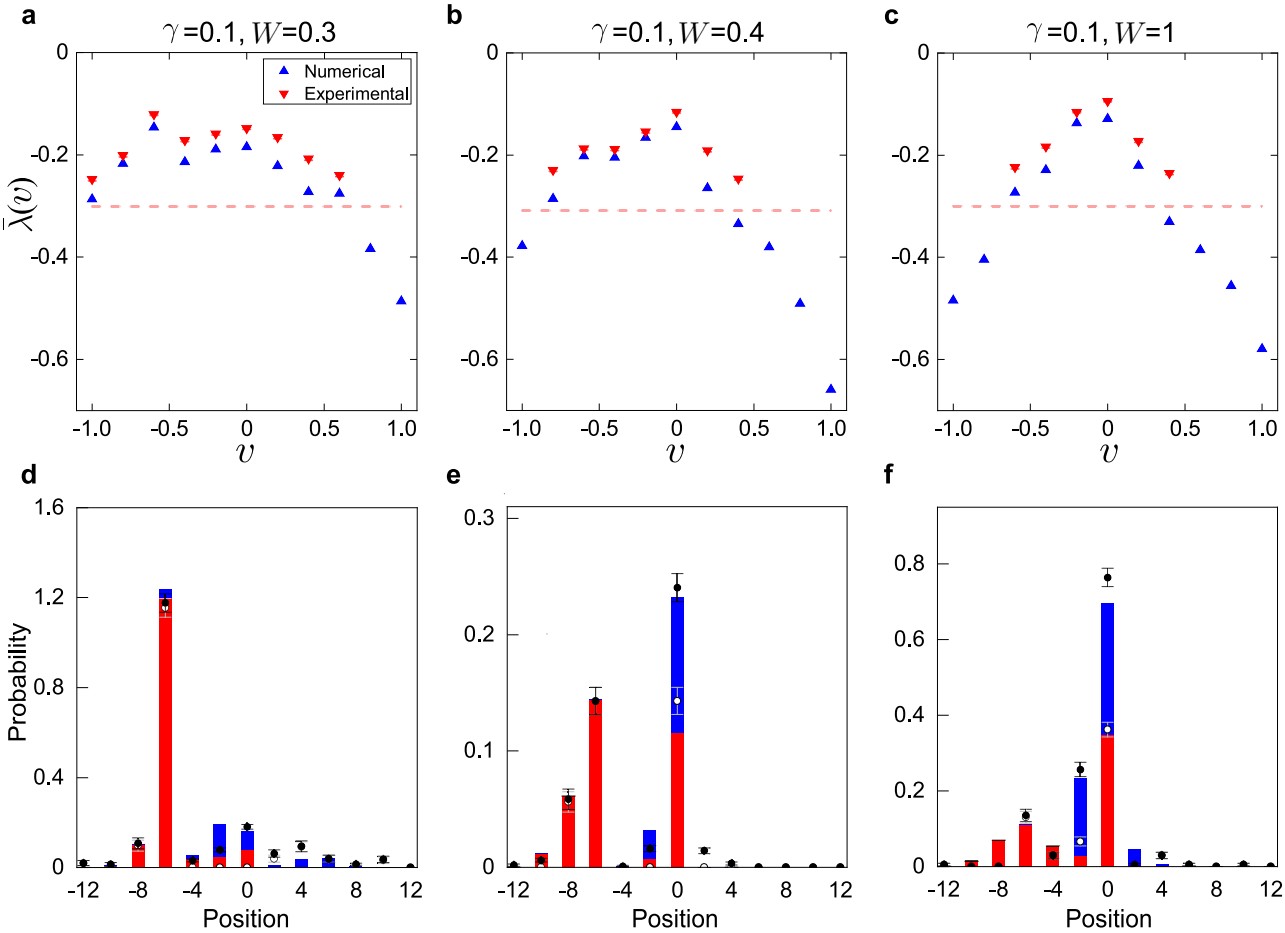

**Fig. 3 Competition between the non-Hermitian skin effect and Anderson localization. a–c** The measured $\bar{\lambda}(v)$ with increasing $W$, under the parameters $\theta_1 = 4.3$, $\theta_2 = 2.175$, and $\gamma = 0.1$. Note that the system is topologically non-trivial under the parameters of **a** and **c**, and is topologically trivial in **b**. **d–f** The experimental data (symbols) and numerical results (bars) for the polarization-resolved photon distribution after the final step $t = 10$. In **a–c** we average over 20 disorder configurations. The symbols used are the same as those in Fig. 2. Error bars are due to the statistical uncertainty in photon-number-counting.

(solid black curve in Fig. 4a)[37], suggesting a unique non-Hermitian criticality. From our observation in previous sections, a non-Hermitian topological Anderson insulator with persistent signatures of the non-Hermitian skin effect is expected in the yellow region of Fig. 4a with $W \lesssim 0.4$, where the disorder has not become dominant.

Here we focus on the impact of the disorder on the topological phase boundary, which we experimentally probe through the time and disorder-averaged biorthogonal chiral displacement, defined for a $t$-step quantum walk as[26,37]

$$\bar{C} = \frac{1}{N} \sum_{n=1}^{N} \sum_{t'=1}^{t} \frac{1}{t} \langle \chi_n(t') | \Gamma X | \psi_n(t') \rangle, \quad (3)$$

where $|\psi_n(t)\rangle = U^t |\psi(0)\rangle$ and $|\chi_n(t)\rangle = \left[ (U^{-1})^\dagger \right]^t |\psi(0)\rangle$, $|\psi(0)\rangle = |0\rangle \otimes |V\rangle$, the subscript $n$ indicates the $n$th disorder configuration (with a total of $N$ configurations), and $X$ is the position operator. Experimentally, we prepare $|\psi_n(t)\rangle$ and $|\chi_n(t)\rangle$ by separately evolving the initial state with $U$ and $(U^{-1})^\dagger$, followed by state tomography to reconstruct $|\psi_n(t)\rangle$ and $|\chi_n(t)\rangle$, respectively, before calculating $\bar{C}$ according to Eq. (3).

In Fig. 4b, we plot the measured $\bar{C}$. Similar to ref. [26], while the measured chiral displacement varies smoothly across the topological phase boundaries due to the limited number of time steps amenable to our experiment, it does show a tendency

consistent with the theoretically predicted phase boundaries. Numerically, it is found that $\bar{C}$ approaches the topological invariants given by the local marker (dashed line) at much larger time steps. The measured $\bar{C}$ is insensitive to $\gamma$, consistent with theoretical predictions using the local marker.

To provide direct evidence for the topological nature of the non-Hermitian topological Anderson insulating state, in Fig. 4c, d, we show the spatial probability distributions following ten-step quantum-walk dynamics close to a domain-wall configuration, where the left ($x \leq -1$) and right ($x \geq 0$) regions feature different parameters ($\theta_2$ in our experiment). When the two regions belong to different topological phases, the time-evolved probability shows a prominent peak at the boundary, indicating the presence of topological edge states (Fig. 4c). This is in sharp contrast to Fig. 4d, where both regions are in the same topological phase. Note that to minimize the impact of the non-Hermitian skin effect, we choose a parameter regime where the non-Hermitian skin effect leads to a directional probability flow through the boundary (corresponding to the probability peaks in the region $x \leq -1$ in Fig. 4c, d), such that the probability accumulation at the boundary in Fig. 4c is unambiguously associated with edge states.

**Discussion.** We report the first experimental observation of a non-Hermitian topological Anderson insulator, achieved by introducing disorder to a discrete-time non-unitary quantum

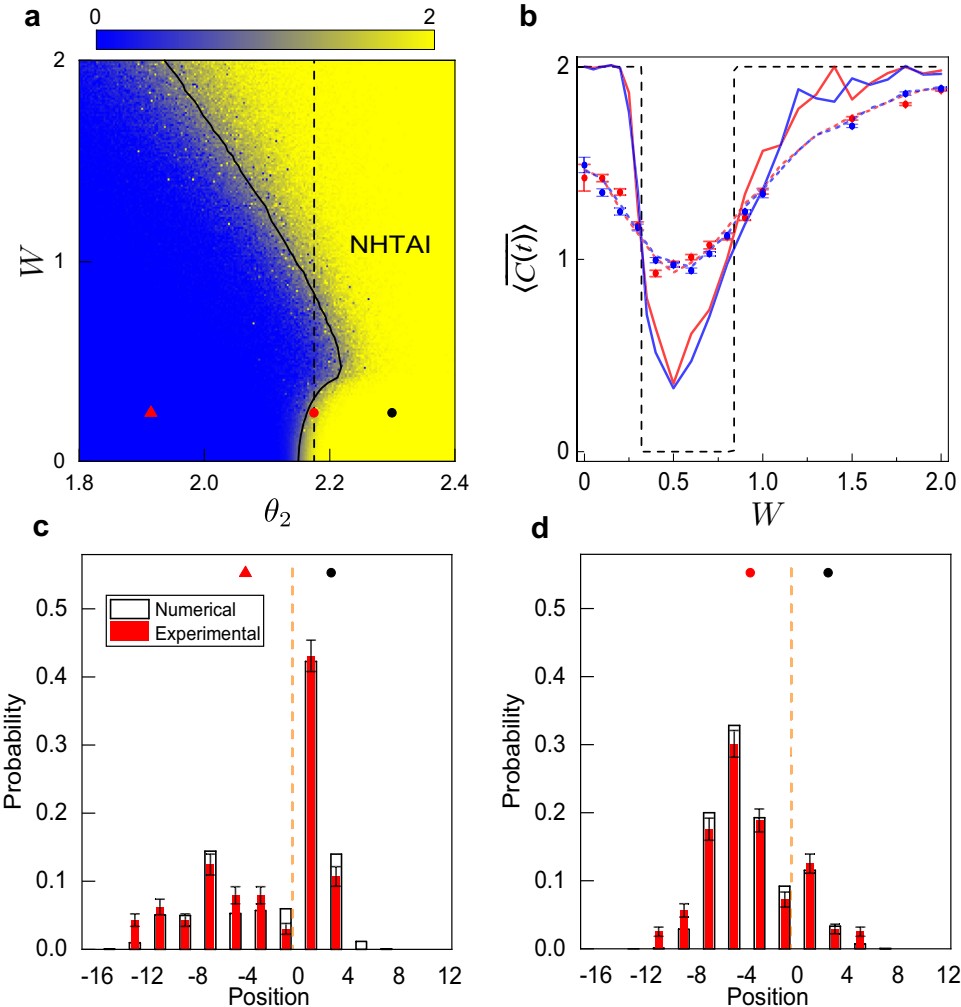

**Fig. 4 Characterizing topology. a** Theoretical phase diagram in terms of the color contour of the numerically evaluated biorthogonal local marker, with $\theta_1 = 4.3 + \delta\theta(x)$, and $\gamma = 0.1$. The yellow (blue) region corresponds to the topologically non-trivial (trivial) phase, thus the non-Hermitian topological Anderson insulator state corresponds to the yellow region with $W > 0$ (labeled NHTAI in **a**). **b** Measured averaged chiral displacement for 9-step quantum walks with $\theta_2 = 2.175$ (vertical dashed line in **a**), averaged over ten different configurations of $\delta\theta(x)$, with $\delta\theta(x)$ taking random values within the range $[-W, W]$. Experimental data were represented by blue and red dots for $\gamma = 0$ and $\gamma = 0.1$, respectively. Error bars are due to the statistical uncertainty in photon-number-counting. Blue and red dashed lines are numerically evaluated chiral displacements for nine-step quantum walks, averaged over 2000 random-disorder configurations, for $\gamma = 0$ and $\gamma = 0.1$, respectively. Blue and red solid lines are numerically evaluated chiral displacement for 400-step quantum walks, averaged over 200 random-disorder configurations, for $\gamma = 0$ and $\gamma = 0.1$, respectively. For all cases, the initial state is $|0\rangle \otimes |V\rangle$. **c, d** are the measured probability distributions after the last time step of ten-step quantum walks close to a boundary (indicated by the vertical dashed lines). In **c**, we set $\theta_2 = 1.915$ for $x \leq -1$ (red triangle in **a**), and $\theta_2 = 2.3$ for $x \geq 0$ (black dot in **a**). In **d**, $\theta_2 = 2.175$ for $x \leq -1$ (red dot in **a**), and $\theta_2 = 2.3$ for $x \geq 0$ (black dot in **a**). In **c, d**, red (black) bars correspond to experimental measurements (numerical simulations), both averaged over ten disorder configurations. The walker is initialized at $|1\rangle \otimes |V\rangle$, and we fix $W = 0.25$.

walk with topology and non-Hermitian skin effect. Using dynamic observables, we demonstrate the two competing localization mechanisms inherent in the system and reveal a disorder-induced topological phase transition. Our experiment lays the foundation for interesting theoretical questions as to the fate of localized states in a non-Hermitian many-body system with skin effect, as well as the interplay of non-Hermiticity, disorder, and many-body interactions therein. On the application side, disorder and non-Hermiticity provide convenient control over key properties of non-Hermitian Anderson insulators, opening routes toward the design of a tunable optical device for engineered quantum transport.

For future studies, it is hopeful to further increase the evolution time of the quantum-walk dynamics based on the time-multiplexed configuration, such that a more accurate determination of the

Lyapunov exponent can be achieved. It would also be interesting to explore similar competitions for higher dimensional non-Hermitian topological Anderson insulators.

## Methods

**Experimental setup**. To implement quantum walks governed by the Floquet operator $U$ in Eq. (1), we adopt a time-multiplexed configuration, encoding the internal coin-state degrees of freedom in the photonic polarization, and the external spatial modes in the discretized temporal shift within a time step[40]. The overall experimental configuration is illustrated in Fig. 1.

The wave packets of photons are generated by a pulsed laser source with a central wavelength of 808 nm, a pulse width of 88 ps, and a repetition rate of 31.25 kHz. The pulses are attenuated to the single-photon level using neutral density filters at the detection stage. For a unitary quantum walk, the probability that a photon undergoes a full round-trip without getting lost or detected is about 0.59 per step and the detection efficiency is 0.03 per step (taking into account the efficiency of APDs and the reflectivity of BSs). We ensure the average photon

number per pulse at the detection stage to be less than $2 \times 10^{-4}$ so that there is a negligible probability of a multi-photon event.

To implement $U$ with a fiber loop configuration, we rewrite the $t$-step time-evolution operator as $U^t = e^{2\gamma t}U_E^t$, where

$$U_E^t = \left[R(\theta_2)M_E SR(\theta_1)M_E SR(\theta_2)\right]^t \quad (4)$$
$$= R(\theta_2)U_{\text{loop}}^t R(-\theta_2),$$

and

$$U_{\text{loop}} = M_E SR(\theta_1)M_E SR(2\theta_2). \quad (5)$$

Here the coin operator $R(\theta_i)$ and the shift operator $S$ are the same as those in Eq. (1) and the ensuing discussions. The polarization-dependent loss operator $M_E = \sum_x |x\rangle\langle x| \otimes (|H\rangle\langle H| + e^{-2\gamma}|V\rangle\langle V|)$, which is related to $M$ through $M = e^\gamma M_E$. For each cycle in the interferometric network, the walker state is subject to the operation $M_E SR(\theta)$, where $\theta$ is alternately modulated to be $2\theta_2$ or $\theta_1$ for odd or even cycles. As such, one cycle in the network roughly corresponds to a half step of the quantum walk. The coin operators $R(\theta_2)$ and $R(-\theta_2)$ are implemented at the input and out-coupling stage, respectively.

More specifically, the operator $R(-\theta_2)$ [$R(\theta_2)$] in Eq. (1) is implemented using two half-wave plates (HWPs) with setting angles $-\theta_2/2$ ($\theta_2/2$) and 0, respectively, before (after) the photon is sent into (coupled out of) the network. For the input, photons are reflected by a low-reflectivity BS with a reflectivity of 5%, such that there is a 5% probability to couple a photon into the network. The same BS is subsequently used as the out-coupler, where photons, after completing cycles in the interferometer, have a 5% probability of being reflected out of the cycle and into the detection module.

Within each interferometer cycle, the photon is first sent through a sandwich-type, QWP(0)-EOM($4\theta_2$)-QWP(90°) configuration[42], which is used to implement the coin operator $R(2\theta_2)$ or $R(\theta_1)$ in Eq. (5). Here QWP is the abbreviation for quarter-wave plates. The birefringent crystal inside the EOM is set at 45° to the $x/y$ axis so that the EOM acts on the photon polarization as $\tilde{R}_{\text{EOM}}(\vartheta) = \begin{pmatrix} 1 & 1 \\ -1 & 1 \end{pmatrix}$ $\begin{pmatrix} e^{i\frac{\vartheta}{2}} & 0 \\ 0 & e^{-i\frac{\vartheta}{2}} \end{pmatrix} \begin{pmatrix} 1 & -1 \\ 1 & 1 \end{pmatrix} = \begin{pmatrix} \cos\frac{\vartheta}{2} & i\sin\frac{\vartheta}{2} \\ i\sin\frac{\vartheta}{2} & \cos\frac{\vartheta}{2} \end{pmatrix}$. The properties impose that $\phi_V(x)/\phi_H(x) = -1$. Thus, in combination with a pair of wave plates, an EOM can be used to modify the polarization of each pulse individually, providing the basis for realizing position-dependent coin operations $R(\vartheta) = \begin{pmatrix} 1 & 0 \\ 0 & -i \end{pmatrix} \tilde{R}_{\text{EOM}}(2\vartheta) \begin{pmatrix} 1 & 0 \\ 0 & i \end{pmatrix} = \begin{pmatrix} \cos\vartheta & -\sin\vartheta \\ \sin\vartheta & \cos\vartheta \end{pmatrix}$. For a disorder-free quantum walk, we sequence the EOM such that $\vartheta = 2\theta_2$ for odd cycles and $\vartheta = \theta_1$ for even cycles.

The shift operator $S$ is implemented by separating different polarization components of a photon using polarizing beam splitters (PBSs) and routing them through fibers of different lengths to introduce a well-defined time delay in between. Specifically, horizontally polarized photons traverse the fiber loop in 751.680 ns, while vertical ones take 33.046 ns longer to complete the trip. The resulting temporal difference corresponds to a step in the spatial domain of $x \pm 1$. As such, each position in each time step is represented by a unique discrete-time bin, i.e., the position information is mapped into the time domain.

To implement the loss operator $M_E$, a pair of HWPs are inserted into each fiber loop, one at the entrance and one near the exit. Since the operator $M_E$ induces a loss in the polarization state $|V\rangle$ with a probability $1 - e^{-4\gamma}$, we adjust the setting angles of the HWPs, such that only the desired components are reflected (transmitted) by the PBS at the exit of the short (long) fiber loop into the blocker, rendering the dynamic within the main cycle non-unitary. We, therefore, read out the evolved states from our experiment with $M_E$ by adding a factor $e^{\gamma t}$.

At the output of the shift operator, the two paths are coherently recombined, and photons are sent back to the input BS for the next split-step. In order to realize a full time step, two cycles in the interferometer network are required, with the setting angle of the EOM alternating between $2\theta_2$ (odd cycle) and $\theta_1$ (even cycle). We introduce static disorder to the coin operator $R(\theta_1)$ for odd cycles. This is achieved by modulating the setting of EOM by a small random amount $\delta\theta \in [-W, W]$ around $\theta_1$. Here $\delta\theta$ is position-dependent but time-independent. Such static disorder preserves the chiral symmetry of $U$.

Finally, after a photon has completed multiple cycles and is coupled out of the network by the BS (with a probability of 5%), the coin operator $R(\theta_2)$ is applied, and the photon registers a click at an APD with a time jitter 350 ps for detection.

**State tomography**. For the detection of the time-averaged chiral displacement, we reconstruct the final state $|\psi(t)\rangle = U^t|\psi(0)\rangle$ and its left vector $|\chi(t)\rangle = \left[(U^{-1})^\dagger\right]^t |\psi(0)\rangle$ for each time step. Here we take the reconstruction of $|\psi(t)\rangle$ as an example. Since $U$ and the initial state $|\psi(0)\rangle = |0\rangle \otimes |V\rangle$ are purely real in the polarization basis $\{|H\rangle, |V\rangle\}$, we have the expansion

$$|\psi(t)\rangle = \sum_x \left[p_H(t,x)|x\rangle \otimes |H\rangle + p_V(t,x)|x\rangle \otimes |V\rangle\right], \quad (6)$$

where the coefficients $p_\mu(t,x)$ ($\mu = H, V$) are also real. Based on these, we perform three distinct measurements $M_i$ ($i = 1, 2, 3$) to reconstruct $|\psi(t)\rangle$ in the basis

$\{|H\rangle, |V\rangle\}$. This amounts to measuring the absolute values and the r signs of the real coefficients $p_\mu(t,x)$, as we detail in the following.

First, we measure the absolute values $\left|p_\mu(t,x)\right|$. After the $t$th time step, photons in position $x$ are sent to a detection unit $M_1$, which consists of PBS and APDs. $M_1$ applies a projective measurement of the observable $\sigma_z$ on the polarization of photons. The counts of the horizontally polarized photons $N_H(t,x)$ and vertically polarized ones $N_V(t,x)$ are registered by the coincidences between one of the APDs in the detection unit, and the APD for the trigger photon. The measured probability distributions are

$$P_\mu(t,x) = \frac{e^{2\gamma}c(t)N_\mu(t,x)}{\sum_x [N_H(t,x) + N_V(t,x)]}, \quad (7)$$

where $c(t) = \text{Tr}\left[U_E^t |\psi(0)\rangle\langle\psi(0)|(U_E^\dagger)^t\right]$. The square root of the probability distribution $P_\mu(t,x)$ corresponds to $\left|p_\mu(t,x)\right|$.

Second, we determine the relative sign between the amplitudes $p_H(t,x)$ and $p_V(t,x)$ via the detection unit $M_2$, which consists of an HWP at 22.5°, a PBS, and APDs. The only difference between $M_2$ and $M_1$ is the HWP at 22.5°, i.e., a projective measurement of the observable $\sigma_x$ on the polarization components of photons. The difference between the probability distributions of the horizontally and vertically polarized photons is given by

$$P_H(t,x) - P_V(t,x) = 2p_H(t,x)p_V(t,x), \quad (8)$$

which determines the relative sign between $p_H(t,x)$ and $p_V(t,x)$.

Third, we probe the relative sign between the amplitudes $p_H(t,x)$ and $p_V(t,x')$, which is necessary to calculate the summation of wave functions in different positions at each time step. We take the relative sign between the amplitudes in the positions $x$ and $x - 2$ as an example. To this end, a detection unit $M_3$ is introduced, consisting of an extra loop, an HWP at 22.5°, a PBS, and APDs. In the extra loop, the EOM is set to realize a rotation $R(\theta_2 + 3\pi/4)$. The horizontally polarized photons at both $x$ and $x - 2$ are combined at the end of the loop. The projective measurement of the observable $\sigma_x$ is applied to the polarization components of photons via an HWP at 22.5°, a PBS, and APDs. The difference between the probability distributions of the horizontally and vertically polarized photons is given by

$$P_H(t,x) - P_V(t,x) = -[p_H(t,x) + p_V(t,x)] \times [p_H(t,x-2) - p_V(t,x-2)]. \quad (9)$$

As we have determined the relative sign between $p_H(t,x)$ and $p_V(t,x)$ [between $p_H(t,x-2)$ and $p_V(t,x-2)$] with $M_2$, we determine, using $M3$, the relative sign between $p_\mu(t,x)$ and $p_\mu(t,x-2)$ for arbitrary $x$.

Note that, as the purpose of reconstructing the final state is to calculate the expectation value of the averaged chiral displacement, the global sign of $p_\mu(x,t)$ is unimportant.

**Biorthogonal local marker and chiral displacement**. Following refs. [26,37], the biorthogonal local marker is defined as

$$\nu(m) = \frac{1}{4}\sum_s \langle m,s|Q\Gamma[X,Q]|m,s\rangle + h.c., \quad (10)$$

where $|m,s\rangle$ is the sublattice state $s$ of the $m$th unit cell, and $X$ is the unit-cell position operator. The biorthogonal projection operator $Q = P_+ - P_-$, with $P_\pm = \sum_n |\phi_\pm^{(n)}\rangle\langle\chi_\pm^{(n)}|$. Where $|\phi_\pm^{(n)}\rangle$ is the $n$th right eigenstate of $U$, satisfying $U|\phi_\pm^{(n)}\rangle = \lambda_\pm^{(n)}|\phi_\pm^{(n)}\rangle$; and $\langle\chi_\pm^{(n)}|$ is the $n$th left eigenstate, with $U^\dagger|\chi_\pm^{(n)}\rangle = \lambda_{n,\pm}^*|\chi_\pm^{(n)}\rangle$. Here $\lambda_{n,+}$ ($\lambda_{n,-}$) lies in the lower (upper) half of the complex plane. Similar to the analysis in refs. [26,37], the biorthogonal local marker serves as the topological invariant in a disordered system, and is reflected in the disorder- and time-averaged chiral displacement defined in Eq. (3).

## Data availability

All other data, any related experimental background information not mentioned in the text, and other findings of this study are available from the corresponding author upon reasonable request. Source data are provided with this paper.

## Code availability

Any simulation and computational codes for this study are available from the corresponding authors upon reasonable request.

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

## Acknowledgements

This work has been supported by the National Natural Science Foundation of China (Grant Nos. 12025401, U1930402, 11974331, and 12088101). W.Y. acknowledges support from the National Key Research and Development Program of China (Grant Nos. 2016YFA0301700 and 2017YFA0304100). L.X. acknowledges support from the Project Funded by China Postdoctoral Science Foundation (Grant Nos. 2020M680006 and 2021T140045).

## Author contributions

Q.L. performed the experiments with contributions from K.W. and L.X. W.Y. developed the theoretical aspects and performed the theoretical analysis with contributions from T.L. and wrote part of the paper. P.X. supervised the project, designed the experiments, analyzed the results, and wrote part of the paper.

## Competing interests

The authors declare no competing interests.
