## [Peer review file · Nature Communications]

REVIEWER COMMENTS

Reviewer #1 (Remarks to the Author):

The authors claim that they observe a non-Hermitian topological Anderson insulator by quantum dynamics. However, I am not persuaded due to the following reasons.

I cannot recommend the publication of this manuscript.

1)

The quantization of the topological number in Fig.4b is not enough. Especially, the experimental data is not quantized at all. In addition, there is no transition at $W=0.3$ and 0.85 as shown in the dotted line. We cannot conclude that the system is topological from Fig.4b.

2)

The authors should assign the phases in Fig.4a. Is the yellow region a topological Anderson insulator?

I am confused that the blue and yellow phases exist even at $W=0$ which means that there is no disorder. Then, the blue and yellow phases cannot be a topological Anderson insulator.

3)

Is the non-Hermitian skin phase non-topological? I understand that the non-Hermitian skin phase also has a topological number. The authors should clarify on this point.

4)

For the skin effect, all of the amplitude including the bulk modes must be localized. However, the peak in Fig.3d does not take at the edge but exist in the bulk site. This contradicts to the skin effect.

5)

Do the non-Hermitian skin effect and the topological Anderson localization coexist?

Or does the skin effect disappear if we increase the disorder?

If so, what is the reason for the disappearance of the skin effect?

6)

Usually, the skin effect is due to the nonreciprocity. Is it also the case for the present system? The authors should clarify the origin of the skin effect.

7)

The red bar is not symmetric in Fig.2c although $\gamma = 0$.

What is the reason of this asymmetry?

8)

The authors should show an intuitive reason that $\lambda(\nu)$ is a good signal for the non-Hermitian skin effect. Only the citation is not enough.

9)

The non-Hermitian skin effect does not occur for the bulk but only occurs in a finite system. On the other hand, the authors show the Lyapunov exponent from bulk dynamics in Fig.2. Is it a result for a finite system?

10)

It is necessary to compare the periodic and open conditions in order to check the existence of the non-Hermitian skin effect. The authors should explicitly show experimental results on them.

11)

In the context of the bulk-edge correspondence, it is not enough to show that a topological number is quantized but we need to show an emergence of topological edge states. The authors should experimentally show the presence of topological edge states in the topological Anderson phase.

12)

The authors should show a topological phase diagram as a function γ and W . This is also the case for Fig.4a.

13)

Why is the phase diagram plotted as a function of θ_2 and W ?

What is the dependence of θ_1 ?

What is the meaning of θ_2 ?

14)

The authors should show a bare data for Fig.4 without taking 200times average of the disorder. The authors should show error bars in Fig.4.

15)

Many important figures are included in supplement information, which makes hard to read the manuscript. The authors should transfer this manuscript elsewhere so that the SI can be included in the main text.

16)

The definition of the winding number (S8) is not correct. The authors should use a winding number for the normalized "d".

17)

What is the relation between Fig.4 and Fig.S5. Is it possible to analytically determine the topological phase boundary in Fig.4?

18)

The transition in Fig.S4 is not sharp and it is hard to determine the phase transition. The authors should improve the result so that the transition is manifest.

19)

The authors should add explanation why Eq.1 is a Floquet operator.

20)

The authors should add more explanation on the experimental realization of Eq.1.

Reviewer #2 (Remarks to the Author):

In the manuscript, the authors experimentally use disordered and non-unitary quantum walks of single photons to simulate a 1D chiral system with the recently-proposed non-Hermitian topological Anderson insulators. The spatial modes are encoded in the discrete arrival time of photons and the hopping between neighbor spatial modes are given by the quantum walks with tunable non-Hermiticity (gain-and-loss) parameter and position-dependent disorder. The authors first measure the Lyapunov exponent to characterize localization in bulk dynamics due to the non-Hermitian skin effect, which is not measured in their previous work (Ref. [29]). Using this dynamic observable, they further study the competition between Anderson localization induced by random disorder and the non-Hermitian skin effect and reveal the resulting non-monotonous localization. Finally, using biorthogonal chiral displacement and state tomography, the authors are able to detect the disorder-induced topological phase transition with the unique biorthogonal criticality. The experimental results are consistent with those by numerical simulations and the theoretical predictions.

In my opinion, the manuscript is well organized and nicely written. This work reveals several interesting phenomena arising from the interplay of disorder, non-Hermiticity, and topology. It is timely, novel, and experimentally impressive, given the current broad interest in non-Hermitian physics and topological states of matter. To the best of my knowledge, it is the first experimental observation of non-Hermitian topological Anderson insulators. Therefore, I believe the work warrants publication in Nature Communications after clarifying the following issues:

1. Can the authors give the effective non-Hermitian and disordered Hamiltonian corresponding to the non-unitary time evolution of random quantum walks? This would be beneficial for understanding the mapping from the quantum dynamics to the simulated system of non-Hermitian topological Anderson insulators.
2. The authors show the horizontal dashed lines in Figs. 2 and 3 as the threshold values below which experimental data are no longer reliable due to photon loss. From the results, the photon loss seems dependent on the shift velocity and disorder strength. I think the authors should explain this issue that is closely related to the experimental data.
3. The local marker and topological invariant in Fig. 4a should be further clarified. Which unit-cell (site) of the local marker is used here? What is the relation between the local marker and the topological invariant (the winding number)? Why the values of the local marker 0 is for topological phase, and 2 for

trivial phase? It is known that the 1D topological and trivial insulators respectively take the non-zero and zero winding numbers.

Reply to Report of Reviewer 1

We thank the reviewer for his/her valuable time, stimulating comments and the comprehensive list of questions. In the following, let us address these questions one-by-one.

The authors claim that they observe a non-Hermitian topological Anderson insulator by quantum dynamics. However, I am not persuaded due to the following reasons. I cannot recommend the publication of this manuscript.

It is regrettable that the reviewer cannot recommend the publication of our manuscript. We believe, however, all of the reviewer's questions can be addressed. In the following, please find a detailed response to the reviewer's questions.

1) The quantization of the topological number in Fig. 4b is not enough. Especially, the experimental data is not quantized at all. In addition, there is no transition at $W = 0.3$ and 0.85 as shown in the dotted line. We cannot conclude that the system is topological from Fig. 4b.

We agree with the reviewer that the measured biorthogonal chiral displacement is not ideally quantized. However, this is the case with all existing measurements with state-of-the-art quantum simulators: direct measurements of topological invariants are rarely exactly quantized under realistic conditions (particularly under disorder). This is the case, for instance, with quantum spin Hall insulators [c.f. *Science* **318**, 766 (2007); *Science*, **325**, 294 (2009)], wherein the quantization of the measured conductance is poor (due to the inevitable breaking of time-reversal symmetry under typical experimental conditions). This is also the case with a very similar measurement of the chiral displacement in the Anderson topological insulator (the Hermitian counterpart of our experiment) using cold atoms [c.f. *Science* **362**, 929 (2018) which was Ref. [25] and is Ref. [26] in the updated version]. While we believe the situation will certainly improve with the ever growing capability of these quantum simulators, it is also very important for us to demonstrate the outreach of the current state-of-the-art, particularly when novel physics is involved.

Similar to Ref. [26], while the measured \bar{C} only qualitatively agrees with the theoretical prediction, the topological transition can be identified by combining experimental data with theoretical characterization (of the localization length, the chiral displacement, and the

numerical scaling of the chiral displacement). More important, we believe what we have shown in Fig. 4 is really one of the best achievable measurements on this phase transition across a broad band of quantum simulators currently available.

2) The authors should assign the phases in Fig. 4a. Is the yellow region a topological Anderson insulator? I am confused that the blue and yellow phases exist even at $W = 0$ which means that there is no disorder. Then, the blue and yellow phases cannot be a topological Anderson insulator.

We thank the reviewer for the stimulating question. In the yellow region, the local topological marker is quantized at 2, meaning it is topologically nontrivial. In the case without disorder, i.e., $W = 0$, the system is the non-Hermitian topological insulator, where the topological invariant [the winding number defined in Eq. (S16)] agrees with the local marker. In the case of $W > 0$, the system is in the non-Hermitian topological Anderson insulator phase, and the local marker plays the role of the topological invariant (as the winding number is now ill-defined).

We clarify this point in the discussion of Fig. 4a.

3) Is the non-Hermitian skin phase non-topological? I understand that the non-Hermitian skin phase also has a topological number. The authors should clarify on this point.

We thank the reviewer for the question and we take the opportunity to clarify this point. The non-Hermitian skin effect has a spectral topological origin, meaning the eigenspectrum of the system under periodic boundary condition features loop structures on the complex plane (hence a spectral winding number can be defined). However, this is distinct from the band topology as in topological insulators, which is the focus here. Also, the spectral winding number is defined differently from that in Eq. (S16) here. A detailed discussion on the spectral topology of non-Hermitian skin effect can also be found in the previous literatures cited in our work as Refs. [17,18].

We add clarification in the revised manuscript and hope the reviewer is satisfied.

4) For the skin effect, all of the amplitude including the bulk modes must be localized. However, the peak in Fig. 3d does not take at the edge but exist in the bulk site. This contradicts to the skin effect.

The non-Hermitian skin effect by definition is the accumulation of population at boundaries. Its origin however, lies in a persistent bulk current, directly reflected in the measured Lyapunov exponent in Figs. 2 and 3. For instance, the measured Lyapunov exponent in Fig. 3a peaks at a finite shift velocity, which corresponds to a shift of the peak in the population profile after some time of evolution in Fig. 3d. This behavior is driven by non-Hermitian skin effect, in contrast to those in Fig. 2a or Fig. 3c where the peak population remains symmetric to the initial site, and the polarization-summed population peaks at the initial site.

To directly demonstrate the presence of non-Hermitian skin effect, in the revised manuscript, we add new experiments to explicitly demonstrate the accumulation of population near boundaries under the non-Hermitian skin effect (see Fig. S8).

5) Do the non-Hermitian skin effect and the topological Anderson localization coexist? Or does the skin effect disappear if we increase the disorder? If so, what is the reason for the disappearance of the skin effect?

The non-Hermitian skin effect and the Anderson localization can coexist and compete with each other. This is explicitly shown and discussed in Figs. 3a, b, where the two peaks (one at $v \sim 0$, one at a finite v) in the Lyapunov exponent compete. In Fig. 3c, we also show what happens at large disorder. Here the Anderson localization becomes so strong that the non-Hermitian skin effect is no longer observable. More specifically, combining Figs. 3 and 4, the non-Hermitian skin effect should still be observable in the yellow region of Fig. 4a, with $W \lesssim 0.4$.

In the revised manuscript, we add some discussions to clarify this point.

6) Usually, the skin effect is due to the nonreciprocity. Is it also the case for the present system? The authors should clarify the origin of the skin effect.

In the original non-Hermitian lattice models in Ref. [8], the non-Hermitian skin effect is directly due to the non-reciprocal hopping. However, non-reciprocal hopping is not the only source of non-Hermitian skin effect. As discussed in [Phys. Rev. Lett. **121**, 086803 (2018); arXiv:2111.04196], on-site loss can also induce non-Hermitian skin effect, as they only differ by a spin rotation (or sublattice state rotation) from models with non-reciprocal hopping. This is the case with our setup as well, where the non-Hermitian skin effect derives from

the interplay of an on-site, polarization-dependent (internal-state dependent) gain and loss (M operator) and an effective coupling between the polarization and the lattice degrees of freedom (S operator).

We add a brief discussion on this point in the revised manuscript.

7) The red bar is not symmetric in Fig. 2c although $\gamma = 0$. What is the reason of this asymmetry?

As the reviewer rightly points out, neither the red nor the blue bars in Fig. 2c are symmetric. This is due to the asymmetry of the initial states and the measurements: the initial states are $|0\rangle \otimes |H\rangle$ and $|0\rangle \otimes |V\rangle$ for the blue (on top) and red (at bottom) bars, while only the horizontally and vertically polarized photons are detected for the blue and red bars, respectively. However, the polarization-summed probability (i.e., the theoretical predictions are indicated by the sum of the red and blue bars, while experimental data are represented by black dots) is symmetric. The polarization-summed probability peaks at the initial site, indicating the absence of the non-Hermitian skin effect.

We now add some discussions in the caption of Fig. 2 to clarify this point.

8) The authors should show an intuitive reason that $\lambda(v)$ is a good signal for the non-Hermitian skin effect. Only the citation is not enough.

By definition, $\lambda(v)$ reflects of the peak of the population distribution at time t . For $\lambda(v)$ to peak at v at a given time t , the time-evolved wave function must peak near $x = vt$. As such, a nonzero v indicates a finite bulk current, which is the origin of the non-Hermitian skin effect. Note that the bulk current can be intuitively understood in terms of the nonreciprocity, meaning the flow in the bulk is directionally biased. It exists in the bulk, thus offering a bulk detection scheme of the non-Hermitian skin effect. This is the reason why $\lambda(v)$ is a reliable bulk signal, widely used for the detection of the non-Hermitian skin effect.

We now add further clarifications in the main text.

9) The non-Hermitian skin effect does not occur for the bulk but only occurs in a finite system. On the other hand, the authors show the Lyapunov exponent from bulk dynamics in Fig. 2. Is it a result for a finite system?

The non-Hermitian skin effect directly manifests itself in the presence of boundaries. However, one can tell from bulk dynamics, as is the case here, whether a system shows non-Hermitian skin effect when a boundary is present. In this sense, the non-Hermitian skin effect is not just for finite systems, but has observable effects on the bulk dynamics and in the thermodynamic limit. Another relevant example can be found in Ref. [34], where a parity-time transition under open boundaries (related to non-Hermitian skin effect) can be measured from bulk dynamics.

In the revised manuscript, we further clarify this point.

10) It is necessary to compare the periodic and open conditions in order to check the existence of the non-Hermitian skin effect. The authors should explicitly show experimental results on them.

We thank the reviewer for his/her stimulating comment. We have fully complied with the reviewer's suggestion and performed additional experiments to directly demonstrate the existence of non-Hermitian skin effects in the presence of boundaries.

As demonstrated in Fig. S8, we consider a domain-wall geometry of non-unitary quantum walks, involving regions with fixed coin parameters $\theta_1 = 4$ and $\theta_2 = 1.63$ but different loss parameters, e.g., γ_L and γ_R for the left ($x < 0$) and right ($x \geq 0$) regions, respectively. The walker starts from the bulk position $x = 4$ with the initial coin state $|V\rangle$. First, we choose $\gamma_L = \gamma_R = 0.1$ and $W = 0$, where there is no boundary. The probability distribution shows a directional flow (Figs. S8a and d), which is the origin of the non-Hermitian skin effect. Second, we choose $-\gamma_L = \gamma_R = 0.1$ and $W = 0$. The walker, initialized near $x = 4$, becomes localized at the domain-wall boundary due to the non-Hermitian skin effect, in sharp contrast to the first case (Figs. S8b and e). Note that since the coin parameters are the same for the two regions, the localization is driven only by the non-Hermitian skin effect but not related to the topological edge states. Finally, we choose $-\gamma_L = \gamma_R = 0.1$ and $W = 1$. The probability peaks at the initial site $x = 4$ as the Anderson localization completely suppresses the probability flow in the bulk (Figs. S8c and f). This is a direct evidence for the competition between the disorder-induced Anderson localization and the non-Hermitian skin effect. The new results are consistent with our bulk measurements of the Lyapunov exponent.

11) In the context of the bulk-edge correspondence, it is not enough to show that a topological number is quantized but we need to show an emergence of topological edge states. The authors should experimentally show the presence of topological edge states in the topological Anderson phase.

We thank the reviewer for the question. However, we are not claiming an observation of bulk-boundary correspondence here, which is surely an interesting topic for future studies, particularly in the context of disorder. This is indeed a challenging open question, since even for Hermitian topological Anderson insulators [c.f. *Science* **362**, 929 (2018); *Nature* **560**, 461 (2018)], the question of bulk-boundary correspondence is still elusive, and a direct measurement of topological edge state is yet to be demonstrated.

12) The authors should show a topological phase diagram as a function γ and W . This is also the case for Fig. 4a.

As we have discussed in the main text, for our particular model, the topological phase boundary is not affected by γ , therefore a phase diagram on the plane of γ — W would be trivial. In the revised manuscript, we provide a phase diagram using θ_1 and W for completeness.

13) Why is the phase diagram plotted as a function of θ_2 and W ? What is the dependence of θ_1 ? What is the meaning of θ_2 ?

As we discuss under Eq. (1), θ_1 and θ_2 are both parameters of the rotation operator R , which are realized using quarter wave plates and the electro-optic modulator in our experiment (please see Methods for details). Our choice of θ_2 for the phase diagram is for convenience, since the alternative choice would not qualitatively change the main physics we wish to discuss.

In the revised manuscript, we provide a phase diagram using θ_1 and W for completeness (see Fig. S5b).

14) The authors should show a bare data for Fig. 4 without taking 200 times average of the disorder. The authors should show error bars in Fig. 4.

We now provide a typical set of raw data in the Supplemental Material. The error bars for experimental measurements are already shown in Fig. 4b.

15) Many important figures are included in supplement information, which makes hard to read the manuscript. The authors should transfer this manuscript elsewhere so that the SI can be included in the main text.

We believe all the key figures are already shown in the main text, which provide a self-sufficient characterization of the non-Hermitian topological Anderson phase. Those in the Supplemental Material are only of secondary importance. Indeed, this is the practice of all *Nature Communications* manuscripts.

16) The definition of the winding number (S8) is not correct. The authors should use a winding number for the normalized “ \mathbf{d} ”.

The definition of the winding number [now Eq. (S16) in the revised version] is correct as is. The Bloch vector \mathbf{d} is already normalized, as discussed immediately below Eq. (S16). We further emphasize this point in the revised manuscript.

17) What is the relation between Fig. 4 and Fig. S5. Is it possible to analytically determine the topological phase boundary in Fig. 4?

Figure S5a shows the topological phase diagram characterized by the non-Bloch winding number under the open boundary condition. No disorder is considered in Fig. S5, so the $W = 0$ axis of Fig. 4a is but a line in Fig. S5a, which we illustrate in the revised manuscript. An analytic derivation of the phase boundary in Fig. 4a is unattainable, due to the complexity of the quantum-walk model.

18) The transition in Fig. S4 is not sharp and it is hard to determine the phase transition. The authors should improve the result so that the transition is manifest.

Actually, there is no phase transition in Fig. S4. The non-monotonic behavior of the blue curve just shows the competition between the non-Hermitian skin effect and the Anderson localization.

19) The authors should add explanation why Eq. 1 is a Floquet operator.

We thank the reviewer for the suggestion. We clarify this point in the revised version. Briefly speaking, U is a Floquet operator because it is repeated acted on the walker state during the time evolution, such that the time-evolved state is $|\psi(t)\rangle = U^t|\psi(0)\rangle$ with t taking integer values.

20) The authors should add more explanation on the experimental realization of Eq. 1.

We believe the experimental realization of Eq. (1) is already fully discussed both in the main text and in the Methods section.

In summary, we have fully taken the reviewers comments into account in the revised manuscript, by adding new experiments, revising figures and presentation. We hope the reviewer is satisfied with the improvement. We believe our work is now suitable for publication in *Nature Communications*.

Reply to Report of Reviewer 2

We thank Reviewer 2 for his/her careful reading of our manuscript and for recommending our work for publication in *Nature Communications*. Following the reviewer's insightful comments and helpful suggestions, we have revised the manuscript accordingly. Below let us address the reviewer's comments point-by-point.

In the manuscript, the authors experimentally use disordered and non-unitary quantum walks of single photons to simulate a 1D chiral system with the recently-proposed non-Hermitian topological Anderson insulators. The spatial modes are encoded in the discrete arrival time of photons and the hopping between neighbor spatial modes are given by the quantum walks with tunable non-Hermiticity (gain-and-loss) parameter and position-dependent disorder. The authors first measure the Lyapunov exponent to characterize localization in bulk dynamics due to the non-Hermitian skin effect, which is not measured in their previous work (Ref. [29]). Using this dynamic observable, they further study the competition between Anderson localization induced by random disorder and the non-Hermitian skin effect and reveal the resulting non-monotonous localization. Finally, using biorthogonal chiral displacement and state tomography, the authors are able to detect the disorder-induced topological phase transition with the unique biorthogonal criticality. The experimental results are consistent with those by numerical simulations and the theoretical predictions.

We agree with the reviewer's summary of our work.

In my opinion, the manuscript is well organized and nicely written. This work reveals several interesting phenomena arising from the interplay of disorder, non-Hermiticity, and topology. It is timely, novel, and experimentally impressive, given the current broad interest in non-Hermitian physics and topological states of matter. To the best of my knowledge, it is the first experimental observation of non-Hermitian topological Anderson insulators. Therefore, I believe the work warrants publication in *Nature Communications* after clarifying the following issues:

We are pleased that the reviewer find our work well-written and our results novel, and we thank the reviewer for recommending our work for publication. We are also glad with the reviewer's endorsement that our experiment is the first observation of non-Hermitian topological Anderson insulator, which we believe would be of interest to a wide audience. In the following, we address the issues that reviewer raised point-by-point.

1. Can the authors give the effective non-Hermitian and disordered Hamiltonian corresponding to the non-unitary time evolution of random quantum walks? This would be beneficial for understanding the mapping from the quantum dynamics to the simulated system of non-Hermitian topological Anderson insulators.

We thank the reviewer for the suggestion. For discrete-time quantum walks, the Floquet operator U is more relevant to the experimental implementation and is formally more compact. We therefore choose to focus on U in the main text. The simulated system can be qualitatively understood as adding disorder to the Floquet operator with non-Hermitian skin effect, similar to the recipe of generating non-Hermitian topological Anderson insulators in Refs. [36,37]. While the non-Hermitian skin effect in Refs. [36,37] is directly due to the non-reciprocal hopping, in our setup, it derives from the interplay of an on-site, polarization-dependent (internal-state dependent) gain and loss (M operator) and an effective coupling between the polarization and the lattice degrees of freedom (S operator). Since the on-site gain and loss is related to a non-reciprocal hopping by an internal-state rotation, the two schemes are closely related (c.f. discussions in arXiv:2111.04196).

Following the reviewer's suggestion, in the Supplemental Material, we now provide a formal expression of the effective non-Hermitian Hamiltonian. We also add discussions concerning the analysis above in the main text.

2. The authors show the horizontal dashed lines in Figs. 2 and 3 as the threshold values below which experimental data are no longer reliable due to photon loss. From the results, the photon loss seems dependent on the shift velocity and disorder strength. I think the authors should explain this issue that is closely related to the experimental data.

We thank the reviewer for offering the opportune to clarify this point. The photon loss is actually not dependent on the shift velocity nor the disorder strength. The photon

loss is caused by the experimental setup. Even for a unitary quantum walk, our single-loop round-trip efficiency is about 0.59. This is estimated by multiplying the efficiency of each individual element in the round trip: the collection efficiency from free space to fiber (~ 0.70), transmission rates of a beam splitter (~ 0.95), that of an EOM (~ 0.95), those of all the other optical elements (~ 0.93). We therefore have $0.59 \simeq 0.70 \times 0.95 \times 0.95 \times 0.93$.

The horizontal dashed lines in Figs. 2 and 3 indicate the event in which at most one photon (one or less one) is detected by an APD placed at some position x . Then the Lyapunov exponent becomes $\lambda_{\min}(v) = \frac{1}{t} \log \sqrt{\frac{1}{N}}$, where N is the total photon number after the t -th step. Experimentally the total photon number N is limited by the single-loop round-trip efficiency. If the values are below the dashed lines, we are not able to reconstruct them from the measured probabilities (or photon numbers). Thus, we emphasize in the main text that “the horizontal dashed lines in Figs. 2 and 3 as the threshold values below which experimental data are no longer reliable due to photon loss.”

We add more discussions on this in the revised Supplemental Material.

3. The local marker and topological invariant in Fig. 4a should be further clarified. Which unit-cell (site) of the local marker is used here? What is the relation between the local marker and the topological invariant (the winding number)? Why the values of the local marker 0 is for topological phase, and 2 for trivial phase? It is known that the 1D topological and trivial insulators respectively take the non-zero and zero winding numbers.

For the local marker in Fig. 4a, we use a unit cell in the bulk, far away from any boundaries. The calculated local marker should coincide with the winding number in the absence of disorder, and serves as the topological invariant in the presence of disorder. Therefore, in the phase diagram Fig. 4a, the blue region has a local marker 0, which is topologically trivial; and the yellow region has a local marker 2, which is topologically non-trivial. The measured biorthogonal chiral displacements qualitatively agree with this conclusion.

In the revised manuscript, we clarify how the local marker is calculated, as well as the topological nature of different regions on the phase diagram. We also move the definition of the local marker to the Methods section.

List of changes

1. We perform new experiments demonstrating signatures of the non-Hermitian skin effect in the presence of boundaries. Please see Fig. S8 and corresponding discussions.
2. We add clarifications of the Floquet operator U on page 2 of the main text.
3. We add brief discussions on the origin of non-Hermitian skin effect in our system on page 3 of the main text.
4. We now add further clarifications on the Lyapunov exponent on page 3 of the main text.
5. We add further clarifications in the caption of Fig. 2.
6. We add brief discussion on the detection of non-Hermitian skin effect from bulk dynamics on page 4 of the main text.
7. We explicitly identify topologically trivial and non-trivial regimes, the regime for the non-Hermitian topological Anderson insulator on the phase diagram.
8. We briefly discuss the coexistence of the topological Anderson insulator and the non-Hermitian skin effect on page 4.
9. We move the definition of the local topological marker to the Methods section.
10. A phase diagram on the θ_1 - W plane is added to Fig. S5.
11. A formal derivation of the effective Hamiltonian is given in the revised Supplemental Material.
12. Typical raw numerical data are given in Table I of the Supplemental Materials per request of the reviewer.
13. We add discussions in the Supplemental Material analyzing photon loss.

REVIEWER COMMENTS

Reviewer #1 (Remarks to the Author):

I am not persuaded by the reply of the authors that this manuscript is suitable for Nature Communications. The main reason is that it is not convincing that the non-Hermitian topological Anderson insulator is actually observed. Indeed, the quantization is too poor both in theoretically and experimentally in their work. The authors should improve the quantization, which should be numerically evaluated by adding more data points for W . Although the authors claim that low quality experimental data are ubiquitous in the field of topological Anderson insulator, the quality is too poor to make any conclusion. I should emphasize that I cannot see the transition at all in their data. In the case that the quantization of the topological number is not good, an alternative is to observe topological edge states only for the topological phase. However, this is not the case in this paper.

There are some comments on the revised manuscript, where the replies are insufficient.

1)

It is not clear how the non-Hermitian topological Anderson insulator and the skin state coexist. The authors should show an intuitive picture of the state for it by showing the spatial profile of the state.

2)

The authors should assign topological phases in Fig.4a.

3)

The author claims "To directly demonstrate the presence of non-Hermitian skin effect, in the revised manuscript, we add new experiments to explicitly demonstrate the accumulation of population near boundaries under the non-Hermitian skin effect (see Fig. S8)."

With respect to this claim, the peak should be taken at the edge. However, this is not the case for Fig.S8. It simply indicates that it is not a skin state.

4)

The authors should assign the names of the phases in Fig.2 and 3.

Reviewer #2 (Remarks to the Author):

The authors have carefully improved their manuscript and supplementary material by adding more discussions (further clarifications) and experimental data. With all the questions and comments being well addressed, I think that the resubmitted manuscript is ready for publication in Nature Communications.

Reviewer #3 (Remarks to the Author):

In this work, the authors realized a time-multiplexed non-unitary quantum walk in their experiment. They observed skin effect in their experiment without disorder. The Lyapunov exponent is measured to illustrate the skin effect. When disorder appears, the authors observed the probability localized in some positions with step. The Lyapunov exponent is also measured to show the competition between the non-Hermitian skin effect and Anderson localization. At the end of experiment, the authors measured the chiral displacement to characterize topology.

After reading through the whole text, I think the experimental realization of the non-unitary quantum walk is convincing. The experiment results about the probability and the Lyapunov exponent are also correct.

By referring to two reviewers' comments, I think the concerns raised by Reviewer #1 are reasonable. The authors need to deal with these concerns.

Reply to Report of Reviewer 1

We thank Reviewer 1 for the comments. In the following, let us address these concerns one-by-one.

I am not persuaded by the reply of the authors that this manuscript is suitable for *Nature Communications*. The main reason is that it is not convincing that the non-Hermitian topological Anderson insulator is actually observed. Indeed, the quantization is too poor both in theoretically and experimentally in their work. The authors should improve the quantization, which should be numerically evaluated by adding more data points for W . Although the authors claim that low quality experimental data are ubiquitous in the field of topological Anderson insulator, the quality is too poor to make any conclusion. I should emphasize that I cannot see the transition at all in their data. In the case that the quantization of the topological number is not good, an alternative is to observe topological edge states only for the topological phase. However, this is not the case in this paper.

In the revised manuscript, we perform new experiments to add more data points in Fig. 4**b**. The new experimental results show that the measured biorthogonal displacement is asymptotically approaching a quantized value of 2 as W increases, which is consistent with numerical results for larger time steps. These new data points help to visualize the transition better.

More importantly, following the reviewer's suggestion, we have also designed and carried out new experiments to demonstrate the presence of topological edge states in a domain-wall configuration (see Figs. 4**c** and **d**), where the left ($x \leq -1$) and right ($x \geq 0$) regions feature different parameters (θ_2 in our experiment)

Specifically, as shown in Fig. 4**c**, when the left ($x \leq -1$) and right ($x \geq 0$) regions belong to different topological phases, the time-evolved probability shows a prominent peak at the boundary, indicating the presence of topological edge states (Fig. 4**c**). This is in sharp contrast to Fig. 4**d**, where both regions are in the same topological phase, and no edge-state signals are observed.

Note that to minimize the impact of non-Hermitian skin effect, we carefully choose the parameters such that the non-Hermitian skin effect leads to a directional probability flow

through the boundary. This results in the probability peaks to the left of the boundary in Figs. 4c and d. It follows that the probability accumulation at the boundary is unambiguously associated with the topological edge states.

We emphasize that, under a domain-wall configuration, the accumulation of the eigenstate wavefunctions at the boundary does not necessarily mean the dynamic accumulation of probabilities in quantum walks. Depending on the parameters, the dynamics at the boundary under the non-Hermitian skin effect can be either of the two cases: a directional probability flow through the boundary (as is the case in Figs. 4c and d); and the accumulation of probability at the boundary (as in Figs. S9b and e, see also our response to point 3 below).

As we have emphasized in the last reply, direct measurements of topological invariants are rarely exactly quantized under realistic conditions, across all existing experimental platforms. This is the case, for instance, with quantum spin Hall insulators [c.f. Science **318**, 766 (2007); Science, **325**, 294 (2009)], wherein the quantization of the measured conductance is poor, due to the inevitable breaking of time-reversal symmetry under typical experimental conditions. This is also the case with measuring the chiral displacement in the topological Anderson insulator using cold atoms [c.f. Science **362**, 929 (2018)], where the data quality, in terms of quantization, is similar to ours.

Nevertheless, we have made an effort in this resubmission to demonstrate the asymptotically quantized behaviour of the biorthogonal displacement at large disorder strengths. Furthermore, as an alternative solution suggested by the reviewer, we are now able to detect the topological edge states, which, to our knowledge, is the first measurement of its kind for topological Anderson insulators (either Hermitian or non-Hermitian). Combined with other numerical evidence such as the localization length shown in Fig. S7, we believe our experimental results are among the best achievable on this phase transition across a broad band of quantum simulators currently available. We hope the reviewer will be satisfied by our new experimental results inspired by his/her suggestions.

1) It is not clear how the non-Hermitian topological Anderson insulator and the skin state coexist. The authors should show an intuitive picture of the state for it by showing the spatial profile of the state.

In the revised Supplemental Material, we add a new figure (Fig. S6) illustrating numerically calculated spatial profiles of the eigen wavefunctions with increasing disorder.

The competition between the Anderson localization and non-Hermitian skin effect is clearly shown, consistent with our experimental data.

Specifically, in the absence of disorder (Fig. S6a), the non-Hermitian skin effect is reflected in the accumulation of eigenstates at the boundary. With a finite W , the non-Hermitian skin effect is still present, but some eigenstates start to show isolated probability humps in the bulk, suggesting the competition between localization and the non-Hermitian skin effect (Fig. S6b). Finally, when W is sufficiently large, the non-Hermitian skin effect is completely suppressed, all eigen wavefunctions become localized and isolated in the bulk (Fig. S6c). Such a competing picture is consistent with our experimentally measured growth rate in Fig. 3 of the main text.

We hope the reviewer will be satisfied by our clarification.

2) The authors should assign topological phases in Fig. 4a.

We have already defined different phases in the figure caption. In the revised version, we now label the non-Hermitian topological Anderson insulator phase in the yellow region.

3) The author claims “To directly demonstrate the presence of non-Hermitian skin effect, in the revised manuscript, we add new experiments to explicitly demonstrate the accumulation of population near boundaries under the non-Hermitian skin effect (see Fig. S8).” With respect to this claim, the peak should be taken at the edge. However, this is not the case for Fig. S8. It simply indicates that it is not a skin state.

In Fig. S9 (Fig. S8 of the previous version), three experiments have been performed. For all cases, the walker is initialized at $x = 4$. In Figs. S9 a and d, no boundary is implemented (i.e., a homogeneous quantum walk), and the probability distribution shows a directional flow (Figs. S9a and d), which is the origin of the non-Hermitian skin effect. In Figs. S9b and e, a domain-wall boundary (an edge) is implemented near $x = 0$. The walker becomes *localized at the boundary* under the non-Hermitian skin effect, in sharp contrast to the first case (which shows a directional flow). This is the direct evidence of non-Hermitian skin effects that the reviewer was asking for.

On the other hand, in Figs. S9c and f, the disorder is sufficiently large that the walker becomes localized near its initial position $x = 4$, which is a direct manifestation of the impact of disorder on the non-Hermitian skin effect.

Furthermore, we note that under non-Hermitian skin effects, eigenstates are exponentially localized at the boundary, but to different extent (state-dependent). It therefore does not necessarily translate to a sharp dynamic localization at the boundary in the quantum-walk dynamics. Depending on the parameters, the dynamics near the domain-wall boundary under the non-Hermitian skin effect can be either one of the two cases: a directional probability flow through the boundary (as is the case in Figs. 4**c** and **d**); and the accumulation of probability near the boundary (as in Figs. S9**b** and **e**).

We also note that our experimental results in Fig. S9 are consistent with a related work [Science **368**, 311 (2020)], in which a similar setup was used to indicate the non-Hermitian skin effect, with qualitatively similar observations.

In this resubmission, we have revised Fig. S9 to make the above discussions more prominent. We show probabilities of longer time evolutions in Figs. S9**a**, **b**, **c** (up to $t = 50$), and we mark the boundary at $x = -0.5$ with a vertical dashed line to help differentiate the boundary localization in Fig. S9**b** from the initial-position localization in Fig. S9**c**).

We hope the discussion and improvements above help to clarify any remaining misunderstanding.

4) The authors should assign the names of the phases in Fig. 2 and 3.

We now add a sentence each in the captions of Figs. 2 and 3, to clarify the topological property of each subplots. We emphasize that the focus of these figures are the existence of non-Hermitian skin effect (Fig. 2) and the competition between the non-Hermitian skin effect with disorder (Fig. 3), rather than topological phase transitions. We therefore opt to minimize the discussions on topology here. Further, we emphasize again that there are no phase transitions between an Anderson-localized state and a state with non-Hermitian skin effect.

Reply to Report of Reviewer 2

The authors have carefully improved their manuscript and supplementary material by adding more discussions (further clarifications) and experimental data. With all the questions and comments being well addressed, I think that the resubmitted manuscript is ready for publication in *Nature Communications*.

We are pleased that Reviewer 2 is satisfied with the revised manuscript. We thank the reviewer for his/her time, and for recommending our work for publication.

Reply to Report of Reviewer 3

In this work, the authors realized a time-multiplexed non-unitary quantum walk in their experiment. They observed skin effect in their experiment without disorder. The Lyapunov exponent is measured to illustrate the skin effect. When disorder appears, the authors observed the probability localized in some positions with step. The Lyapunov exponent is also measured to show the competition between the non-Hermitian skin effect and Anderson localization. At the end of experiment, the authors measured the chiral displacement to characterize topology.

After reading through the whole text, I think the experimental realization of the non-unitary quantum walk is convincing. The experiment results about the probability and the Lyapunov exponent are also correct.

We thank Reviewer 3 for carefully reading our manuscript, and for finding our experiment convincing and correct.

By referring to two reviewers comments, I think the concerns raised by Reviewer 1 are reasonable. The authors need to deal with these concerns.

We have made a great effort to address the comments and concerns both in our last response and this. We believe the manuscript is now suitable for publication in *Nature Communications*.

List of changes

1. We added new data points to Fig. 4b to demonstrate the asymptotically quantized behavior of the biorthogonal displacement at large disorder strengths.
2. We performed new experiments demonstrating the presence of topological edge states for the non-Hermitian topological Anderson insulator phase, and revised Fig. 4 accordingly. A paragraph is added in the revised version to discuss these new results. We also add a sentence at the end of introduction on the new experiments.
3. We added a new figure (Fig. S6) in the Supplemental Material to show the competition between non-Hermitian skin effect and disorder.
4. We revised Fig. S9 to further bring out the impact of the non-Hermitian skin effect.
5. We corrected typos throughout the main text and Supplemental Material and improved the writing to enhance the readability of our manuscript.

Reviewer #1 (Remarks to the Author):

I appreciate that the authors have added new experimental data although the quantization of experimental data is still not so good. However, I understand that the quantization is very hard in the current technology. I agree that this paper is worth to be reported in Nature Communications.

Reply to Report of Reviewer 1

I appreciate that the authors have added new experimental data although the quantization of experimental data is still not so good. However, I understand that the quantization is very hard in the current technology. I agree that this paper is worth to be reported in Nature Communications.

We are pleased that Reviewer 1 is satisfied with the revised manuscript. We thank the reviewer for his/her time, and for recommending our work for publication in Nature Communications.